# The Revival of Surgery in Crohn’s Disease—Early Intestinal Resection as a Reasonable Alternative in Localized Ileitis

**DOI:** 10.3390/biomedicines9101317

**Published:** 2021-09-26

**Authors:** Matthias Kelm, Christoph-Thomas Germer, Nicolas Schlegel, Sven Flemming

**Affiliations:** Department of General, Visceral, Transplantation, Vascular and Pediatric Surgery, Center of Operative Medicine (ZOM), University Hospital of Wuerzburg, 97080 Wuerzburg, Germany; Kelm_M@ukw.de (M.K.); Germer_C@ukw.de (C.-T.G.); Schlegel_N@ukw.de (N.S.)

**Keywords:** surgery, Crohn’s disease, terminal ileitis, inflammatory bowel disease, surgical outcome

## Abstract

Crohn’s disease (CD) represents a heterogeneous and complex disease with no curative therapeutic option available to date. Current therapy is mainly antibody-based focusing on the immune system while other treatment alternatives such as surgery are considered to be “last options”. However, medical therapy for CD results in mild to severe side effects in a relevant amount of patients and some patients do not respond to the medication. Following that, quality of life is often significantly reduced in this patient cohort, thus, therapeutic alternatives are urgently needed. Updated evidence has revealed that surgery such as ileocecal resection (ICR) might be a potential therapeutic option in case of localized terminal ileitis since resection at early time points improves quality of life and significantly reduces the postoperative need for immunosuppressive medication with low rates of morbidity. In addition, new surgical approaches such as Kono-S anastomosis or inclusion of the mesentery result in significantly reduced rates of disease recurrence and reoperation. Based on the new evidence, the goal of this review is to provide an update on the role of surgery as a reasonable alternative to medical therapy in the interdisciplinary treatment of patients with CD.

## 1. Introduction

Crohn’s disease (CD) belongs to the group of inflammatory bowel diseases and represents a major socioeconomic burden due to a rising incidence and prevalence globally making it a challenging task for all healthcare systems [1,2,3,4,5]. Usually, CD is diagnosed in young adulthood and occurs as a mucosal or transmural inflammation while sometimes epithelioid granulomas can be found within the gut wall. Most often CD is primarily localized in the terminal ileum (Figure 1). However, it can affect all parts of the gastrointestinal tract from the oral cavity to the anus [6]. According to the current consensus criteria, the extent of CD can be defined as localized or extensive CD. Localized disease is defined as intestinal CD affecting <30 cm in extent, which usually applies to a localization in the ileocecal region but can also be located in colonic segments or the proximal intestinal region. Extensive CD is defined as affecting >100 cm in extent independent of the localization. This definition aims to recognize the inflammatory burden of the disease while it has the disadvantage of being randomly chosen and leaving an undefined “gap” in the range of 30–100 cm [7]. Nonetheless, the definition is important when discussing the individual therapeutic strategy for patients. Despite the relevance of CD and the increasing scientific efforts to understand this complex disease, its pathophysiology still remains unclear and no curative option exists to date [8]. This is in contrast to ulcerative colitis that is restricted to the colon and can therefore be cured surgically by resection of the whole colon. Therefore, patients with CD suffer from a life-long burden of this disease that commonly leads to a dramatic overall reduction in quality of life. Current concepts suggest that CD displays a multifactorial pathogenesis with genetic predisposition, aberrant immune reactions, environmental factors, altered microbiota, and changes of the gut barrier function all together contributing to the onset and perpetuation of the disease. Since there is no curative therapy available for CD, the main therapeutic goal is to induce remission in the short term and to maintain remission in the long term to restore patients’ quality of life. Current therapeutic strategies mainly focus on the aberrant reaction within the immune system using anti-inflammatory and immunosuppressive medication. However, despite the introduction of new medical therapies such as biologicals during recent decades, up to 50% of patients treated with immunosuppressants did not show improvement of clinical symptoms while others suffered from mild to severe side effects resulting in reduced quality of life [9,10]. Common side effects include fatigue, arthralgia, recurrent infections, osteoporosis, or cancer (melanoma and lymphoma) [9,11]. Nevertheless, newer biologicals such as vedolizumab or ustekinumab demonstrate significantly lower rates of side effects, are considered as relatively safe treatment options in CD, and show significantly increased therapeutic efficacy [12,13,14]. However, rates of treatment discontinuation due to lack/loss of response even with newer biologicals remain relevant and patients with a primary nonresponse to antiTNF agents show lower rates of response to second-line biologics [11,14,15]. Following that, the development of new therapeutic strategies to not only decrease side effects but to improve patients’ quality of life as well as treatment efficacy is of great interest. Accordingly, due to the lack of evidence, the option to potentially induce remission in patients by early surgical intervention has not been established in the past and surgery in CD was only seen as therapeutic option in case of enteric complications over a long period of time [16]. However, recent evidence challenges this view, which will most likely induce a paradigm shift in the future, while expanding the role of surgery for CD also raises new questions and challenges which need to be addressed. Thus, the goal of this review is to summarize the current evidence on the interdisciplinary treatment of limited CD, present new surgical strategies, and provide an outlook on future scientific and clinical issues.

## 2. Historic Development of the Role of Surgery to Treat CD

The history of CD and its treatment algorithm is long and includes various revolutionary milestones with its management being in consistent evolution. In 1932, Crohn, Ginzburg, and Oppenheimer were the first to report patients suffering from regional ileitis who were treated at The Mount Sinai Hospital, New York, NY, USA [17]. At that time, primary therapy of patients with regional ileitis consisted of surgical resection of the inflamed gut segments. Based on a rapid evolvement of pathophysiologic concepts, novel medical therapies have been developed resulting in the replacement of surgery as the primary treatment option for patients with CD. This development was also driven by the fear of surgical complications including the need for repeated surgery leading to short bowel syndrome and frequent relapses of the disease. Since then, surgical treatment has only served as an alternative approach in case of complications or refractory disease [16,18]. Most likely this view has led clinical researchers to define the primary focus of studies on CD medications not only on remission of the disease but also as time to avoid and delay surgery. This long-standing paradigm shift to abandon surgery as primary therapeutic option was further supported by randomized-controlled studies demonstrating that the biological-based treatment results in clinical and endoscopical improvement of patients suffering from CD resulting in a decreased need of surgery during a one-year followup [19,20]. However, general rates of surgical resection for patients with CD outside the studies mentioned above have remained stable during the last 20 years with 20–35% during the first five years following initial diagnosis [10,21], despite the “aggressive” use of biologicals (step-down approach) [22,23]. In addition, even more powerful therapies such as biologicals initially resulted in increased rates of side effects and reduced quality of life for patients [9,24]. Based on this background but with newer biologicals demonstrating improved safety profiles and treatment efficacy, it is reasonable that quality of life should be chosen as primary end point for (prospective) studies on therapies for patients suffering from CD in the future instead of surgery-free survival. Therefore, due to the constant rates of surgical resections in CD patients despite the introduction of new medical therapies and major scientific progress, the question about the role of surgery as therapeutic alternative, at least for subgroups of patients, needs be re-evaluated urgently.

## 3. New Evidence Strengthens the Role of Early Surgical Intervention in Limited CD

Today, consensus recommendations about surgery as therapeutic approach in patients with CD remain heterogeneous. For instance, guidelines of the American College of Gastroenterology describe surgery as reserved for severe enteric complications only such as bowel obstruction, abscess formation, perforation, or the presence of medically refractory disease (Table 1) [18].

This perspective with differences in clinical practice result in decreased numbers of CD-related surgery, which is demonstrated by data from Canada and Australia [25,26]. In contrast, the British National Institute for Health and Care Excellence (NICE) recommends surgery already at an early stage of the disease as therapeutic alternative [27]. This approach is also supported by German guidelines whereas British and German societies recommend introducing all therapeutic alternatives to patients while taking into account the state of disease as well as individual risks and benefits and personal preferences [27,28]. Similarly, updated European Crohn’s and Colitis Organization (ECCO) guidelines introduce surgery as a primary treatment option in the case of localized CD as an equal therapeutic alternative to Tumor Necrosis Factor alpha inhibitor infliximab. Importantly, this recommendation includes patients with active inflammation but without stenosis and represents a major adaptation in clinical practice [29]. Those adaptations in guidelines increase the relevance of surgery in CD substantially and are mainly due to new studies in the last few years. These studies demonstrated advantages for early surgical intervention especially in regard to disease recurrence and rates of reoperation. Aratari et al. showed in the postoperative histopathological reappraisal after early surgical intervention in patients with a first-time diagnosis of CD, the disease-free interval was significantly prolonged in comparison to surgery at a later stage of the disease (median: 54.2 months) during a followup of more than 10 years (*p* = 0.02, 95%-CI 0.35–0.92) [30]. In the case of localized disease to the terminal ileum, data from our own patient cohort demonstrated a significantly decreased need for immunosuppressive medication postoperatively for patients who received primary ileocecal resection (ICR) compared to patients who were primarily treated medically and received surgery at a later stage (37.9% vs. 80%, *p* = 0.001) [31]. Importantly, rates of postoperative complications were low and no mortality was seen. The positive effect of surgery on the postoperative follow-up is underlined by studies from Italy and Hungary, which showed that early surgical intervention in patients with CD is associated with a significantly decreased need for immunosuppressive medication and rates of reoperation (10.8 versus 5.8 years, *p* < 0.01) in short- and long-term follow up [32,33]. In an Australian cohort analysis, similar results were observed since early surgical intervention (<6 months after initial diagnosis) resulted in decreased rates of reoperation (14.2% versus 31.3%, *p* = 0.041) and less need for biologicals postoperatively (33.3% versus 60%, *p* = 0.004) [34]. This was confirmed by the retrospective CONNECT study, which showed a significantly decreased need for biologicals after early operation during a follow up of nine years [35]. However, it must also be considered that most of these studies have a retrospective character and patient cohorts are heterogeneous due to the clinical variability of CD.

**Table 1 biomedicines-09-01317-t001:** Recommendation of international guidelines regarding isolated ileocolonic Crohn’s disease.

Society/Organization	Recommendation
American College of Gastroenterology [18]	Surgery is reserved for severe enteric complications only such as bowel obstruction, abscess formation, perforation or the presence of medically refractory disease.
British National Institute for Health and Care Excellence (NICE) [27]	Surgery is recommended at an early stage of the disease as therapeutic alternative to medical therapy.
German Society of Gastroenterology (DGVS) and German Society of Visceral Surgery (DGAV) [28]	Surgery is recommended as primary treatment option in case of localized CD as an equal therapeutic alternative to biological (medical) therapy.
European Crohn’s and Colitis Organization (ECCO) [29]	Surgery is recommended as primary treatment option in case of localized CD as an equal therapeutic alternative to biological (medical) therapy.

Nonetheless, according to those retrospective analyses there might be a potential advantage for early surgical intervention in localized CD regarding the need for antibody-based medication and reoperation. These findings are substantiated by two prospective studies comparing early surgical resection versus medical therapy. Gerdin et al. analyzed patients with isolated CD in the Swedish Crohn Trial [36]. Even though the study was terminated prematurely due to the slow inclusion rate and changes in clinical practice resulting in a small study cohort, patients who received ileocecal resection demonstrated improved quality of life and general health in comparison to patients who were randomized for medical therapy, while no differences were seen for disease activity during follow-up. Furthermore, in 2017, Ponsioen et al. published the LIR!C trial which compared laparoscopic ICR to infliximab therapy in patients with localized terminal ileitis [37]. In this randomized controlled trial, patients were included if localized CD was newly diagnosed and clinical course was refractory to three months of immunosuppressive medication including steroids and nonbiological drugs. After 12 months, patients showed improved quality of life following surgical resection in comparison to infliximab therapy with low morbidity rates. Furthermore, 18% of patients initially receiving medical therapy needed surgery during a follow-up of 12 months. Just recently published long-term data from the LIR!C trial re-evaluated these results. During a median follow-up of 63.5 months, 26% of patients with ICR needed antibody-based therapy postoperatively. Of those without antibody-based therapy, 48% were on immunomodulators. In comparison, 48% of patients with initial infliximab therapy had to be operated due to CD during follow-up. No patient of the infliximab-group was medication free while some of them needed therapy escalation. While the duration of treatment effect was comparable between both groups, a similar amount of patients in each group needed additional treatment during follow-up due to disease activity (approximately 60% in each group) [38]. In addition to the LIR!C trial, Wright et al. demonstrated in a prospective trial that quality of life was significantly improved after surgical resection while the effect was sustained during a follow-up period of 18 months [39].

## 4. Early Surgical Therapy in Crohn’s Disease Is Characterized by Low Morbidity and Mortality

Common arguments against surgery are potential surgical complications and short bowel syndrome due to the need for reoperations. However, several studies demonstrated that surgery in patients with CD is a safe therapeutic approach with very low morbidity and mortality [31,40,41]. These observations are also based on the introduction of bowel sparing strategies such as different techniques of strictureplasty and minimal invasive surgery including endoscopic and laparoscopic approaches. In the last decade, the development of minimal invasive surgery has been dramatically accelerated due to the introduction of single-incision laparoscopic (SILS) and robotic surgery, which significantly improved the quality of surgery regarding perioperative recovery of patients, enhanced body image and cosmesis [42,43,44]. Furthermore, the implementation of structured perioperative patient care programs such as prehabilitation and “Fast Track Surgery”/“ERAS” (enhanced recovery after surgery) results in earlier recovery and decreased morbidity [45,46,47]. These perioperative and surgical developments and advancements help to partially offset the negative impact of patient’s comorbidities and complicated disease phenotype on postoperative outcomes including disease-, surgical- and nonsurgical-related complications.

In general, even if laparoscopic surgery shows similar results regarding disease recurrence, the laparoscopic approach should be preferred since the advantages of minimally-invasive surgical techniques are well established as mentioned above [40,48]. However, due to its complexity, Crohn’s surgery should be performed in high-volume centers only since an interdisciplinary treatment approach is necessary and rates of complications are associated with the center’s and surgeon’s experience [49,50]. This is in part due to increased rates of laparoscopic approaches and improved complication management but also due to regular interdisciplinary interactions between gastroenterologists and surgeons which provides optimal evidence-based and individualized patient care. Adequate planning and risk management of CD-related surgery based on the center’s experience is critical to reduce surgical complications and in-hospital mortality as well as to increase the patient’s quality of life [51,52]. Short bowel syndrome is a complication which needs to be avoided urgently and strategies for bowel preserving surgery such as resecting as little bowel as necessary as well as strictureplasty are well established [29,53]. A further advantage of early surgical intervention is that usually less bowel segments need to be resected, thus, bowel preserving surgery can be performed more easily at early time points of CD. This is especially relevant since rates for reoperation are decreased and disease-free survival prolonged after early surgery as outlined before. In case of surgical intervention at a later time point, extended resections are often necessary since there is an increased risk for complications such as conglomerates, fistulas, or abscesses making the principle of bowel preserving surgery difficult if not even impossible [54].

## 5. Early Surgical Therapy Seems to Be Associated with Decreased Health Care Costs

It is well known that Crohn’s disease as a life-long disease causes relevant health-care costs which are mainly driven by biological therapy [5,55,56,57,58,59]. Furthermore, the introduction of biological-based treatments were associated with increased rates of (severe) side effects resulting in additional health-care costs [60]. Therefore, besides patients’ outcome parameters, health-related cost-effectiveness plays an important role in health care systems worldwide as well. Based on the study population of the LIR!C trial, De Groof et al. demonstrated that mean total direct health care-related costs per patient at one year were significantly lower in the group of patients with surgical resection compared to patients receiving infliximab (mean difference −8931€) [38]. Similarly, the POCER study confirmed that postoperative CD associated costs are primarily medication-related [61]. Therefore, while further studies are needed to compare the cost-effectiveness of other medications than infliximab as well as biosimilars to draw a final conclusion, the initial results demonstrate a potential benefit for surgical resection regarding cost-effectiveness in CD therapy at least in comparison to original biologicals.

## 6. Improved Surgical Technique May Further Support the Beneficial Effects of Surgery to Maintain Remission in CD

Aside from the progress in medical therapy, there has been also a development in surgical techniques for Crohn’s disease resulting in significant improvements for patient outcome [40,62]. While laparoscopic surgery was introduced in the twentieth century, its advancement has resulted in significantly decreased hospitalization rates, complications, and overall morbidity while showing a similar effect on disease outcome compared to open surgery. In general, side-to-side anastomoses are associated with lower rates of postoperative recurrence than end-to-end anastomoses and should therefore be preferred [29,63]. Within the last decade, a new technique for anastomosis in CD was introduced by Kono et al. [64]. This technique (Kono-S anastomosis) is based on the idea that the inflammation in CD originates from the mesentery so the anastomosis should be created as far away as possible from it (antimesenteric handsewn anastomosis) while avoiding devascularization and denervation of the tissue. In addition, other key factors such as a large lumen to avoid fecal stasis and provide functional peristalsis as well as healthy intestinal segments were also included in the concept of Kono et al. Initial studies demonstrate impressive results regarding the risk of surgical recurrence in CD with a surgical recurrence-free survival of more than 95% after 10 years following Kono-S anastomosis [41]. Importantly, while endoscopic recurrence rates were comparable between both groups, additive postoperative application of infliximab significantly decreased the number of surgical recurrences in patients with conventional anastomoses [64]. In addition, a recent meta-analysis revealed a strongly decreased mean Rutgeert score and an endoscopic recurrence of only 5% while rates of overall complications were rare [65]. The first randomized-controlled trial (SuPREMe-CD study) compared Kono-S anastomosis versus a conventional latero-lateral anastomosis and confirmed the significant reduction in postoperative endoscopic and clinical recurrence after a followup of 24 months [66]. Further, international multicenter trials are currently recruiting patients and preliminary results are expected during the upcoming years (NCT03256240).

Moreover, there has been a growing body of evidence that CD might be a primary mesenteropathy in recent years [67,68,69,70,71]. In line with the principle of CD originating from the mesentery, Coffey et al. demonstrated that the inclusion of the mesentery during ICR results in significantly reduced rates of reoperation compared to conventional ICR (40% versus 2.9%, *p* = 0.007) [72]. However, while the two patient cohorts included in this study were small (30 versus 34 patients), patients without resection of the mesentery had increased rates of positive resection margins (79% versus 16%), which might be associated with increased rates of recurrence. Therefore, due to the heterogeneity of both cohorts in this study the conclusion about the role of the mesentery remains open and needs to be further evaluated. To answer the question of whether mesocolic excision during primary ICR reduces postoperative disease recurrence, an international multicenter randomized trial is currently in process [73].

When discussing the role of surgery in inflammatory diseases, Crohn’s disease in particular, positive resection margins and their potential consequences regarding disease recurrence are a highly relevant field but a standardized algorithm is missing. While the evidence regarding resection margins is clearly defined in case of oncological diseases, the consequence of positive resection margins in case of terminal ileitis remains open. Regarding anastomotic leakage, Garofalo et al. reported that a positive proximal resection margin is associated with increased rates of anastomotic leakage postoperatively in comparison to patients without microscopic involvement of inflammatory cells of the anastomosis [74]. However, Schineis et al. and Aaltonen et al. reported in single center studies that no differences regarding anastomotic leakages were seen following microscopic inflammation, and that overall postoperative complications rates were low [75,76]. Similarly, analyzing our own cohort we did not see increased rates of anastomotic leakages for patients with positive resection margins following ileocecal resection (data not published yet). Therefore, due to the heterogeneous evidence, extended resections are currently not recommended but further data are needed to analyze the question in more detail.

A different question, however, is the role of positive resection margins on postoperative disease recurrence. This is especially relevant in cases of early endoscopic and/or clinical recurrence. While the exact pathogenesis for postoperative recurrence is unknown, various risk factors such as smoking, granulomas in the resection specimen, previous intestinal resection, and myenteric plexitis have been identified [77,78,79]. However, the relationship between positive resection margins and early disease recurrence remains unknown and a standard algorithm is missing. Different studies demonstrated a significantly increased risk for endoscopic recurrence following ileocecal resection in case of microscopic inflammation at the resection margins. Following ileocecal resection, Poredska et al. showed that 56.5% of patients with positive resection margins suffered from endoscopic recurrence after six months in comparison to 4.8% with noninflamed resection margins (*p* < 0.001) [80]. Furthermore, another study from France confirmed an increased risk of clinical and surgical recurrence after resection in case of a positive histological margin. During a follow up of five years, 51% of patients with positive margins had disease recurrence while 34% had not (*p* = 0.034). [81] This is underlined by a multicenter study which demonstrated that a transmural lesion, especially at the oral resection margin, is independently associated with an increased risk for postoperative recurrence (75% versus 46%) [82]. Interestingly, Wasman et al. demonstrated in more detail that patients with active inflammation at the colonic resection margin after ICR might have a different and more aggressive disease and might require more intense medical treatment [83]. Furthermore, two recent meta-analyses demonstrated that histological positive resection margins increased clinical and surgical recurrence with a trend towards endoscopic recurrence but definitions of margin positivity and postoperative recurrence varied throughout the studies and limited the final conclusion [78,79]. Despite the presented studies above, the overall evidence remains heterogeneous with various studies showing no effect for resection margins on the postoperative course of disease. For instance, Zemel et al. did not find differences for clinical or endoscopic recurrence on patients with CD following ICR in a retrospective study. While the mean time from surgery to recurrence was comparable between both groups (4.5 versus 4.4 years), 75.5% of patients with a positive resection margin suffered from recurrence compared to 69.9% without microscopic inflammation (*p* = 0.57) [84]. Furthermore, in a single center study from Finland no significant differences were seen regarding the postoperative rates of disease recurrence in case of active inflammation at the resection margin [76]. Since the goal of CD surgery is usually a limited bowel resection to avoid short-bowl syndrome, extended resections are not supported based on the available evidence to date. On the other hand, the role of resection margins in Crohn’s disease remains a double-edged sword because the removal of microscopic inflammation might decrease disease recurrence and the postoperative need for anti-inflammatory or immunosuppressive medication. Therefore, further prospective randomized studies are necessary to clearly determine the role of resections margins in regard of anastomotic leakage and disease recurrence for patients with CD following ileocecal resection.

## 7. Summary and Future Perspectives

Overall, despite the introduction of new medical therapies such as biologicals, the treatment of CD and its complications remains challenging. While initial studies revealed promising results for new types of medical therapy, these highly promising outcomes could not be confirmed by data outside of these studies. Rates of severe side effects were especially high for older biologicals but newer biologicals such as vedolizumab or ustekinumab have been shown to be relatively safe and effective treatment options [9,11,12,13,14]. However, rates of treatment discontinuation remain relevant even with newer biologicals due to lack/loss of response [14,15]. Furthermore, the primary endpoint of studies should not be operation-free survival, but quality of life and remission as revealed by clinical and endoscopy scores. An additional (secondary) aspect may be cost-effectiveness but evaluation for the cost reduction by biosimilars is still lacking. For the parameters of quality of life and cost effectiveness, recent data revealed advantages for early surgical intervention in a subgroup of patients with localized ileocecal CD while demonstrating low rates of postoperative complications [31,37,38,85]. Thus, ICR is a safe therapeutic alternative to medical therapy in this special cohort.

Based on the current evidence, all available therapeutic alternatives (medical versus surgical) should be discussed with patients with localized ileocecal CD early, considering the risks and benefits as well as the personal preferences of patients as already recommended by British (NICE) and European (ECCO) guidelines [27,29]. This is an important and relevant development since surgery was indicated only in case of therapeutic refraction or enteric complications in the past. However, optimal treatment regimens remain controversial and not only medical but also surgical improvements are necessary to improve long-term outcome for patients with CD. A major difficulty for the adequate interpretation and comparison of studies about therapies in patients with CD remains the heterogeneity of patient cohorts as well as the complexity of the disease with multiple confounders. Aside from larger, homogenous prospective studies, new scientific approaches should also be included to optimize the effect of medical therapy and to reduce side effects by thinking outside the box and not only inhibit inflammation by immunosuppressants or antibody-mediated medications but also support mucosal and histological healing with new classes of drugs. In addition, operative strategies and surgical techniques such as the Kono-S anastomosis and minimally-invasive/robotic-assisted surgery needs to be developed further as well. This includes important aspects such as the role of positive resection margins and resection of the mesentery, which will be part of future analysis and might provide the potential to decrease postoperative recurrence rates. Thus, medical and surgical approaches to improve multidisciplinary treatment regimens mainly focusing on the individual patient’s quality of life are essential. Importantly, strategies not only for primary treatment but also for remission prophylaxis need to be evaluated and optimized as well. Due to the complexity and heterogeneity of CD, individual therapeutic regimens should be in the center of future patient-directed therapies (Figure 1). Thus, all disease- and treatment-related side effects including medical and surgical side effects need to be considered to provide optimal and individualized care for patients suffering from CD which can be only achieved by an interdisciplinary team of gastroenterologists, radiologists, pathologists, and surgeons with regular and open-minded communication.

## 8. Conclusions

There has been a renaissance of surgical approaches in the multidisciplinary treatment of isolated Crohn’s disease with improved quality of life and fewer side effects based on new evidence in recent years. Nevertheless, further efforts are still needed to develop and introduce novel medical and surgical therapy options to continuously improve the outcome of patients suffering from Crohn’s disease. To do so, sufficient cooperation between medical and surgical therapeutic approaches is critical.

## Figures and Tables

**Figure 1 biomedicines-09-01317-f001:**
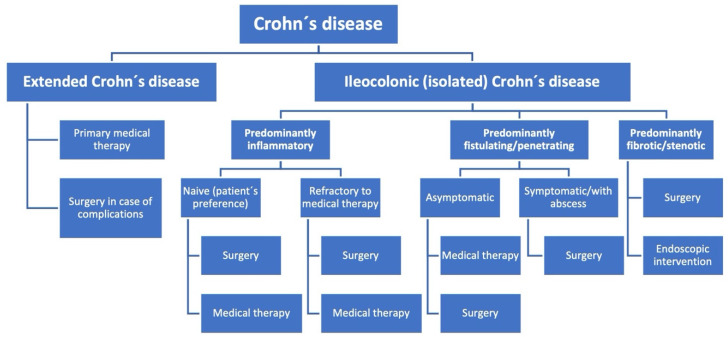
Flow chart for treatment strategies in Crohn’s disease.

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
