# Peer review of "The Revival of Surgery in Crohn’s Disease—Early Intestinal Resection as a Reasonable Alternative in Localized Ileitis"

_biomedicines, 2021, doi:10.3390/biomedicines9101317_

Round 1
Reviewer 1 Report
This is a review article outlining the evidence supporting surgical intervention early in the course of Crohn's disease rather than reserving it only for refractory or complicated cases. This is an important discussion to be had and this is a welcome article to contribute to the debate. However, the data relating to this discussion is difficult to dissect and apply to individual patients with individual prognostic factors, due to a large number of confounders in all studies to date. Hence caution must be applied when expressing opinions and even recommendations based on these data.
General points:
The vibe of the article seems to be very positive for early surgical intervention and stresses the negative aspects of medical interventions fairly excessively. The right balance must be found between a review article and opinion piece, particularly when the evidence is not straight forward, and the applicability of this approach to a particular patient is far from certain.
Specific points:
Line 28-29 - Crohn's is not specifically described to be transmural. It can often be mucosal. Better to desribe more generally. Granulomas are also found only in a minority
Line 44 - rather than saying "usually leads to a dramatic overall reduction" more correct to say "commonly leads to..."
Line 58 - sentence needs to be rewritten
Line 82-86 - true that biologics decrease need for surgery yet the shift away from surgery pre-dated this data by many years
Line 89-91 - This statement is misleading - newer biologics actually seem to be safer and with less frequent side-effects than older treatments. One reference supporting this statement refers to azathioprine, a drug in use for over 50 years and is not considered a biologic or aggressive
Line 159-163 - This statement is also misleading. 26% of patients post ICR were on antiTNF medication. Yet 48% were on immunomodulators. Also important to mention is the duration of therapy effect, ie. the time until a further treatment was required after the initial intervention, which was the same for both groups
Line 210-213 - Must also factor in the impact of biosimilar dramatically reducing the price of biologic therapy.
Line 229-231 - This is also potentially misleading. Recurrence in the Kono study was defined as surgical recurrence. Endoscopic recurrence was present in both groups. Additionally, post operative medications were administered in the study specifically with anti-TNF found to be protective for recurrence. The conclusions described in this paragraph need more accurately portrayed.
The discussion from line 251-305 is important but the relevance is only peripherally related to point of the paper. This should be shortened in order to maintain the focus of the paper.
Line 308-310 needs to be rewritten. Unclear what is being said, and if I understood correctly suggests severe side-effects from newer biologics, which is not necessarily true.
Line 313-316 must take into account cost reduction with biosimilar
More discussion needs to revolve around the difficulties interpreting studies with differing end-points which don't necessarily compare the early-surgery and late-surgery with equal footing with multiple confounders. Also patient selection, with multiple variables such as age, disease phenotype and other risk factors, is an important point to include in the discussion. There is some data, some quite recent, discussing no therapy at all in CD, meaning for isolated mild TI disease, no treatment should also be included as an option. This should also be discussed in analysis of current data, if including only a specific subset of patients thought to be uncomplicated as a potential surgical candidate, what is the prognosis of no treatment as a potential alternative.
Author Response
General points:
The vibe of the article seems to be very positive for early surgical intervention and stresses the negative aspects of medical interventions fairly excessively. The right balance must be found between a review article and opinion piece, particularly when the evidence is not straight forward, and the applicability of this approach to a particular patient is far from certain.
Answer: We thank the reviewer for bringing up this point. The manuscript focuses on the role of surgery in localized Crohn`s Disease and not CD in general. For localized CD, current evidence supports the role of surgery as summarized in our manuscript and as presented in well-established international and national guidelines (ECCO, British, German). Nevertheless, as stated in our manuscript, we do think that the treatment of localized CD needs to be interdisciplinary including medical therapy. Due to the complexity of the disease, a multidisciplinary approach is necessary to continuously improve treatment of CD patients. The goal of our review is the introduction of surgery as additional treatment approach and not to sideline medical interventions, therefore, we edited the manuscript at several parts. Also, critical aspects such as complications and resection margins were already addressed before.
Specific points:
Line 28-29 - Crohn's is not specifically described to be transmural. It can often be mucosal. Better to describe more generally. Granulomas are also found only in a minority
Answer: We have changed this part accordingly reviewer´s suggestion (Line 28-30).
Line 44 - rather than saying "usually leads to a dramatic overall reduction" more correct to say "commonly leads to...“.
Answer: Done (Line 44).
Line 58 - sentence needs to be rewritten
Answer: Thank you for your comment. We have re-written this sentence (Line 57-60).
Line 82-86 - true that biologics decrease need for surgery yet the shift away from surgery pre-dated this data by many years
Answer: This is a valid point, we changed the manuscript to „longstanding paradigm shift“ and „further supported“. (Line 80-81)
Line 89-91 - This statement is misleading - newer biologics actually seem to be safer and with less frequent side-effects than older treatments. One reference supporting this statement refers to azathioprine, a drug in use for over 50 years and is not considered a biologic or aggressive.
Answer: We clarified the sentence and removed the mentioned citation. (Line 87-90).
Line 159-163 - This statement is also misleading. 26% of patients post ICR were on anti-TNF medication. Yet 48% were on immunomodulators. Also important to mention is the duration of therapy effect, ie. the time until a further treatment was required after the initial intervention, which was the same for both groups.
Answer: This is correct and we edited the manuscript. However, the important message of this first prospective randomized trial is, that laparoscopic ileal resection is a reasonable alternative to infliximab.
Line 210-213 - Must also factor in the impact of biosimilar dramatically reducing the price of biologic therapy.
Answer: We added this comment and re-wrote the sentence (Line 217-220).
Line 229-231 - This is also potentially misleading. Recurrence in the Kono study was defined as surgical recurrence. Endoscopic recurrence was present in both groups. Additionally, post operative medications were administered in the study specifically with anti-TNF found to be protective for recurrence. The conclusions described in this paragraph need more accurately portrayed.
Answer: It is correct that recurrence in the initial study by Kono et al. was defined as surgical recurrence and we specified that in the manuscript (Line 237). Regarding postoperative prophylaxis, the majority of patients (64%) did not receive additive postoperative anti-TNF therapy but further evaluation of the effect of anti TNF in this particular study was not done. Thus, in our opinion a specific conclusion about the role of anti TNF in Kono-S anastomosis cannot be drawn based on this study. In addition, as outlined in the follow-up sentence, a recent meta-analysis demonstrated a decreased endoscopic recurrence and mean Rutgeert score.
The discussion from line 251-305 is important but the relevance is only peripherally related to point of the paper. This should be shortened in order to maintain the focus of the paper.
Answer: We thank the reviewer for bringing up this comment but we clearly disagree. From a surgical perspective, resection margins are a highly relevant aspect for short- and long-term disease outcomes and growing clinical and basic research focus on this specific topic in CD. To our mind, the future role of resection margins is at least as important as the Kono-S anastomosis and needs to be addressed in detail when the role of surgery and novel surgical strategies in localized ileitis are discussed. Also, as outlined by the reviewer under „general aspects“, it demonstrates that certain aspects in CD surgery need to be re-evaluated in the future and that there is an ongoing discussion about potential negative aspects of surgery in CD.
Line 308-310 needs to be rewritten. Unclear what is being said, and if I understood correctly suggests severe side-effects from newer biologics, which is not necessarily true.
Answer: The sentence has been re-written and specified (Line 316-317).
Line 313-316 must take into account cost reduction with biosimilar.
Answer: Rephrased (Line 320-321).
More discussion needs to revolve around the difficulties interpreting studies with differing end-points which don't necessarily compare the early-surgery and late-surgery with equal footing with multiple confounders. Also patient selection, with multiple variables such as age, disease phenotype and other risk factors, is an important point to include in the discussion. There is some data, some quite recent, discussing no therapy at all in CD, meaning for isolated mild TI disease, no treatment should also be included as an option. This should also be discussed in analysis of current data, if including only a specific subset of patients thought to be uncomplicated as a potential surgical candidate, what is the prognosis of no treatment as a potential alternative.
Answer: This is a relevant aspect which we addressed in the discussion section (Line 331-339). We agree with the reviewer that the different studies are difficult to compare and that final conclusions need to be drawn carefully. Therefore, we suggest that all available evidence including medical and surgical aspects should be discussed for patients by an interdisciplinary team including gastroenterologists and surgeons since treatment needs to be individualized.

Reviewer 2 Report
Dear Authors, it is a well written, interesting and wide narrative review about the clinical approach to adopt in case of complicated Crohn's disease terminal ileitis. It is prevalently focused on the surgical point of view. I believe the review is up to date and almost all the most currently debated issues (new surgical strategies and anastomoses, ERAS approach, etc....) have been considered.
I have two minor comments:
-figure 1 does not add a lot in this review, in my opinion it could be removed to avoid misleading interpretation in the figure legend.
-At page 5 lines 187-190, it is reported the need for surgical treatment in experienced Centres (quoting two manuscript published in 2008 and 2013): I believe it could be of interest to clarify this aspect, reporting some more recent experiences about what is the different national or continental variations in high/low-volume hospitals (higher rate of laparoscopy in high-volume hospitals, holistic and personalized approach to the patient in presence of multidisciplinary dedicated IBD units, etc)
Author Response
Figure 1 does not add a lot in this review, in my opinion it could be removed to avoid misleading interpretation in the figure legend:
Answer: The figure was removed from the manuscript.
At page 5 lines 187-190, it is reported the need for surgical treatment in experienced Centres (quoting two manuscript published in 2008 and 2013): I believe it could be of interest to clarify this aspect, reporting some more recent experiences about what is the different national or continental variations in high/low-volume hospitals (higher rate of laparoscopy in high-volume hospitals, holistic and personalized approach to the patient in presence of multidisciplinary dedicated IBD units, etc).
Answer: We thank the reviewer for this important comment. We clarified this sentence, provided more detailed information, and included two additional studies. While the exact reasons vary in studies, it seems to be that at least complication management as well as multidisciplinary case discussions are strong indicators for improved patient outcome (Line 190-196).

Round 2
Reviewer 1 Report
I have seen some of the modifications made to the manuscript based on my comments and I do not feel that my concerns have been adequately addressed.
There seems to be an excessive bias on side effects of newer biologicals with no significant references to back up the claim of severe side-effects with newer biologics such as vedolizumab, ustekinumab etc.
The corrections to the descriptions regarding the Liric study or the Kono-s study does not correct the somewhat misleading portrayal of the data.
I recommend more significant rewriting of the manuscript with addition of supporting references as per my original review.
Author Response
Dear Dr. Wang
Dear Reviewer #1,
Thank you for your additional comments on our manuscript entitled “The revival of surgery in Crohn`s Disease – early intestinal resection as reasonable alternative in localized ileitis”.
We value your feedback and comments and edited the manuscript according to your statements. Please see below a point-by-point response to the reviewers’ comments and concerns. All changes in the manuscript are highlighted in the text.
Comments from Reviewer 1:
I have seen some of the modifications made to the manuscript based on my comments and I do not feel that my concerns have been adequately addressed.
There seems to be an excessive bias on side effects of newer biologicals with no significant references to back up the claim of severe side-effects with newer biologics such as vedolizumab, ustekinumab etc.
The corrections to the descriptions regarding the LIR!C study or the Kono-S study does not correct the somewhat misleading portrayal of the data.
I recommend more significant rewriting of the manuscript with addition of supporting references as per my original review.
Answer:
We appreciate the comments of Reviewer 1 and changed the manuscript significantly at several sections to address the mentioned concerns adequately:
- We added five new references (Ref 11-15) and provide more specific background about decreased rates of side effects and improved treatment efficacy for newer biologicals such as vedolizumab and ustekinumab. However, rates of treatment discontinuation remain relevant even with newer biologicals, thus, new therapeutic options (medical and surgical) are necessary (Line 55-64, 96-97, 329-333).
- Regarding the comments about the LIR!C study, we added specific information including rates of medication, duration of therapy and need for additional treatment as suggested by Reviewer 1 originally (Line 162-175).
- Regarding the original comment from Reviewer 1 about the Kono-S study, we clarified that endoscopic recurrence rates were comparable between both groups and that additional postoperative application infliximab therapy significantly decreased the number of surgical recurrences in patients with conventional anastomoses. (Line 247-250)